# Unsupervised Progressive Learning and the STAM Architecture

## Abstract

We first pose the Unsupervised Progressive Learning (UPL) problem: an online representation learning problem in which the learner observes a non-stationary and unlabeled data stream, learning a growing number of features that persist over time even though the data is not stored or replayed. To solve the UPL problem we propose the Self-Taught Associative Memory (STAM) architecture. Layered hierarchies of STAM modules learn based on a combination of online clustering, novelty detection, forgetting outliers, and storing only prototypical features rather than specific examples. We evaluate STAM representations using clustering and classification tasks. While there are no existing learning scenarios that are directly comparable to UPL, we compare the STAM architecture with two recent continual learning models, Memory Aware Synapses (MAS) and Gradient Episodic Memories (GEM), after adapting them in the UPL setting.

## 1 Introduction

The *Continual Learning (CL)* problem is predominantly addressed in the supervised context with the goal being to learn a sequence of tasks without "catastrophic forgetting" (Goodfellow et al., 2013; Parisi et al., 2019; van de Ven & Tolias, 2019). There are several CL variations but a common formulation is that the learner observes a set of examples $\{(x_i, t_i, y_i)\}$, where $x_i$ is a feature vector, $t_i$ is a task identifier, and $y_i$ is the target vector associated with $(x_i, t_i)$ (Chaudhry et al., 2019a;b; Lopez-Paz & Ranzato, 2017). Other CL variations replace task identifiers with task boundaries that are either given (Hsu et al., 2018) or inferred (Zeno et al., 2018). Typically, CL requires that the learner either stores and replays some previously seen examples (Aljundi et al., 2019a;b; Gepperth & Karaoguz, 2017; Hayes et al., 2019; Kemker et al., 2018; Rebuffi et al., 2017) or generates examples of earlier learned tasks (Kemker & Kanan, 2018; Liu et al., 2020; Shin et al., 2017).

The *Unsupervised Feature (or Representation) Learning (FL)* problem, on the other hand, is unsupervised but mostly studied in the *offline context*: given a set of examples $\{x_i\}$, the goal is to learn a *feature vector* $h_i = f(x_i)$ of a given dimensionality that, ideally, makes it easier to identify the explanatory factors of variation behind the data (Bengio et al., 2013), leading to better performance in tasks such as clustering or classification. FL methods differ in the prior $P(h)$ and the loss function. Autoencoders, for instance, aim to learn features of a lower dimensionality than the input that enable a sufficiently good reconstruction (Bengio, 2014; Kingma & Welling, 2013; Tschannen et al., 2018; Zhou et al., 2012). A similar approach is self-supervised methods, which learn representations by optimizing an auxiliary task (Berthelot et al., 2019; Doersch et al., 2015; Gidaris et al., 2018; Kuo et al., 2019; Oord et al., 2018; Sohn et al., 2020).

In this work, we focus on a new and pragmatic problem that adopts some elements of CL and FL but is also different than them – we refer to this problem as *single-pass unsupervised progressive learning* or *UPL* for short. UPL can be described as follows:
1. the data is observed as a non-IID stream (e.g., different portions of the stream may follow different distributions and there may be strong temporal correlations between successive examples),
2. the features should be learned exclusively from unlabeled data,
3. each example is "seen" only once and the unlabeled data are not stored for iterative processing,
4. the number of learned features may need to increase over time, in response to new tasks and/or changes in the data distribution,
5. to avoid catastrophic forgetting, previously learned features need to persist over time, even when the corresponding data are no longer observed in the stream.

The UPL problem is encountered in important AI applications, such as a robot learning new visual features as it explores a time-varying environment. Additionally, we argue that UPL is closer to how animals learn, at least in the case of *perceptual learning* (Goldstone, 1998). We believe that in order to mimic that, ML methods should be able to learn in a streaming manner and in the absence of supervision. Animals do not "save off" labeled examples to train in parallel with unlabeled data, they do not know how many "classes" exist in their environment, and they do not have to replay/dream periodically all their past experiences to avoid forgetting them.

To address the UPL problem, we describe an architecture referred to as STAM ("Self-Taught Associative Memory"). STAM learns features through *online clustering* at a hierarchy of increasing receptive field sizes. We choose online clustering, instead of more complex learning models, because it can be performed through a single pass over the data stream. Further, despite its simplicity, clustering can generate representations that enable better classification performance than more complex FL methods such as sparse-coding or some deep learning methods (Coates et al., 2011; Coates & Ng, 2012). STAM allows the number of clusters to increase over time, driven by a *novelty detection* mechanism. Additionally, STAM includes a brain-inspired *dual-memory hierarchy* (short-term versus long-term) that enables the conservation of previously learned features (to avoid catastrophic forgetting) that have been seen multiple times at the data stream, while forgetting outliers. To the extent of our knowledge, the UPL problem has not been addressed before. The closest prior work is CURL ("Continual Unsupervised Representation Learning") (Rao et al., 2019). CURL however does not consider the single-pass, online learning requirement. We further discuss this difference with CURL in Section 6.

## 2 STAM ARCHITECTURE

In the following, we describe the STAM architecture as a sequence of its major components: a hierarchy of increasing receptive fields, online clustering (centroid learning), novelty detection, and a dual-memory hierarchy that stores prototypical features rather than specific examples. The notation is summarized for convenience in the Supplementary Material (SM) (section SM-A).

**I. Hierarchy of increasing receptive fields:** An input vector $\mathbf{x_t} \in \mathbb{R}^\mathbf{n}$ (an image in all subsequent examples) is analyzed through a hierarchy of $\Lambda$ layers. Instead of neurons or hidden-layer units, each layer consists of STAM units – in its simplest form a STAM unit functions as an online clustering module. Each STAM unit processes one $\rho_l \times \rho_l$ *patch* (e.g. $8 \times 8$ subvector) of the input at the $l$'th layer. The patches are overlapping, with a small stride (set to one pixel in our experiments) to accomplish translation invariance (similar to CNNs). The patch dimension $\rho_l$ increases in higher layers – the idea is that the first layer learns the smallest and most elementary features while the top layer learns the largest and most complex features.

**II. Centroid Learning:** Every patch of each layer is clustered, in an online manner, to a set of centroids. These time-varying centroids form the *features* that the STAM architecture gradually learns at that layer. All STAM units of layer $l$ share the same set of centroids $C_l(t)$ at time $t$ – again for translation invariance.[1] Given the $m$'th input patch $\mathbf{x_{l,m}}$ at layer $l$, the nearest centroid of $C_l$ selected for $\mathbf{x_{l,m}}$ is

$$\mathbf{c_{l,j}} = \arg \min_{c \in C_l} d(\mathbf{x_{l,m}}, \mathbf{c}) \tag{1}$$

where $d(\mathbf{x_{l,m}}, \mathbf{c})$ is the Euclidean distance between the patch $\mathbf{x_{l,m}}$ and centroid $\mathbf{c}$.[2] The selected centroid is updated based on a learning rate parameter $\alpha$, as follows:

$$\mathbf{c_{l,j}} = \alpha \, \mathbf{x_{l,m}} + (1 - \alpha)\mathbf{c_{l,j}}, \quad 0 < \alpha < 1 \tag{2}$$

A higher $\alpha$ value makes the learning process faster but less predictable. A centroid is only updated by at most one patch and the update is not performed if patch is considered "novel" (defined in the next paragraph). We do not use a decreasing value of $\alpha$ because the goal is to keep learning in a non-stationary environment rather than convergence to a stable centroid.

**III. Novelty detection:** When an input patch $\mathbf{x_{l,m}}$ at layer $l$ is significantly different than all centroids at that layer (i.e., its distance to the nearest centroid is a statistical outlier), a new centroid is created

---

[1]We drop the time index $t$ from this point on but it is still implied that the centroids are dynamically learned over time.

[2]We have also experimented with the L1 metric with only minimal differences. Different distance metrics may be more appropriate for other types of data.

in $C_l$ based on $\mathbf{x_{l,m}}$. We refer to this event as *Novelty Detection (ND)*. This function is necessary so that the architecture can learn novel features when the data distribution changes.

To do so, we estimate in an online manner the distance distribution between input patches and their nearest centroid (separately for each layer). The novelty detection threshold at layer $l$ is denoted by $\hat{D}_l$ and it is defined as the 95-th percentile ($\beta = 0.95$) of this distance distribution.

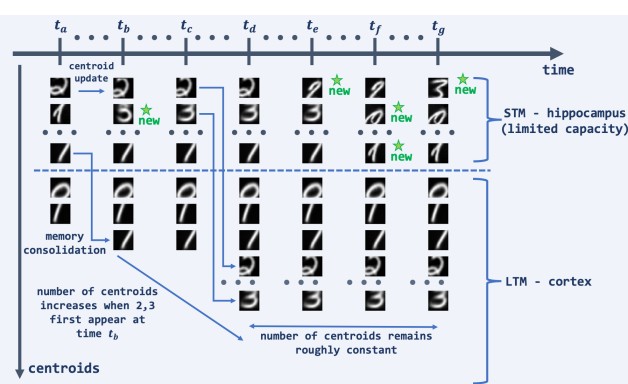

**Figure 1:** A hypothetical pool of STM and LTM centroids visualized at seven time instants. From $t_a$ to $t_b$, a centroid is moved from STM to LTM after it has been selected $\theta$ times. At time $t_b$, unlabeled examples from classes '2' and '3' first appear, triggering novelty detection and new centroids are created in STM. These centroids are moved into LTM by $t_d$. From $t_d$ to $t_g$, the pool of LTM centroids remains the same because no new classes are seen. The pool of STM centroids keeps changing when we receive "outlier" inputs of previously seen classes. Those centroids are later replaced (Least-Recently-Used policy) due to the limited capacity of the STM pool.

**IV. Dual-memory organization:** New centroids are stored temporarily in a *Short-Term Memory (STM)* of limited capacity $\Delta$, separately for each layer. Every time a centroid is selected as the nearest neighbor of an input patch, it is updated based on (2). If an STM centroid $\mathbf{c_{l,j}}$ is selected more than $\theta$ times, it is copied to the *Long-Term Memory (LTM)* for that layer. We refer to this event as *memory consolidation*. The LTM has (practically) unlimited capacity and the learning rate is much smaller (in our experiments the LTM learning rate is set to zero).

This memory organization is inspired by the Complementary Learning Systems framework (Kumaran et al., 2016), where the STM role is played by the hippocampus and the LTM role by the cortex. This dual-memory scheme is necessary to distinguish between infrequently seen patterns that can be forgotten ("outliers"), and new patterns that are frequently seen after they first appear ("novelty").

When the STM pool of centroids at a layer is full, the introduction of a new centroid (created through novelty detection) causes the removal of an earlier centroid. We use the Least-Recently Used (LRU) policy to remove atypical centroids that have not been recently selected by any input. Figure 1 illustrates this dual-memory organization.

**V. Initialization:** We initialize the pool of STM centroids at each layer using randomly sampled patches from the first few images of the unlabeled stream. The initial value of the novelty-detection threshold is calculated based on the distance distribution between each of these initial STM centroids and its nearest centroid.

## 3 CLUSTERING USING STAM

We can use the STAM features in unsupervised tasks, such as offline clustering. For each patch of input $\mathbf{x}$, we compute the nearest LTM centroid. The set of all such centroids, across all patches of $\mathbf{x}$, is denoted by $\Phi(\mathbf{x})$. Given two inputs $\mathbf{x}$ and $\mathbf{y}$, their pairwise distance is the Jaccard distance of $\Phi(\mathbf{x})$ and $\Phi(\mathbf{y})$. Then, given a set of inputs that need to be clustered, and a target number of clusters, we apply a spectral clustering algorithm on the pairwise distances between the set of inputs. We could also use other clustering algorithms, as long as they do not require Euclidean distances.

## 4 CLASSIFICATION USING STAM

Given a small amount of labeled data, STAM representations can also be evaluated with classification tasks. We emphasize that the labeled data is not used for representation learning – it is only used to associate previously learned features with a given set of classes.

**I. Associating centroids with classes:** Suppose we are given some labeled examples $X_L(t)$ from a set of classes $L(t)$ at time $t$. We can use these labeled examples to associate existing LTM centroids at time $t$ (learned strictly from unlabeled data) with the set of classes in $L(t)$.

Given a labeled example of class $k$, suppose that there is a patch $\mathbf{x}$ in that example for which the nearest centroid is $\mathbf{c}$. That patch contributes the following association between centroid $\mathbf{c}$ and class $k$:

$$f_{\mathbf{x},\mathbf{c}}(k) = e^{-d(\mathbf{x},\mathbf{c})/\bar{D}_l} \tag{3}$$

where $\bar{D}_l$ is a normalization constant (calculated as the average distance between input patches and centroids). The *class-association vector* $\mathbf{g_c}$ between centroid $\mathbf{c}$ and any class $k$ is computed aggregating all such associations, across all labeled examples in $X_L$:

$$g_{\mathbf{c}}(k) = \frac{\sum_{\mathbf{x}\in X_L(k)} f_{\mathbf{x},\mathbf{c}}(k)}{\sum_{k'\in L(t)}\sum_{\mathbf{x}\in X_L(k')} f_{\mathbf{x},\mathbf{c}}(k')}, \quad k = 1\dots L(t) \tag{4}$$

where $X_L(k)$ refers to labeled examples belonging to class $k$. Note that $\sum_k g_{\mathbf{c}}(k){=}1$.

**II. Class informative centroids:** If a centroid is associated with only one class $k$ ($g_{\mathbf{c}}(k) = 1$), only labeled examples of that class select that centroid. At the other extreme, if a centroid is equally likely to be selected by examples of any labeled class, ($g_{\mathbf{c}}(k) \approx 1/|L(t)|$), the selection of that centroid does not provide any significant information for the class of the corresponding input. We identify the centroids that are *Class INformative (CIN)* as those that are associated with at least one class significantly more than expected by chance. Specifically, a centroid $\mathbf{c}$ is CIN if

$$\max_{k\in L(t)} g_{\mathbf{c}}(k) > \frac{1}{|L(t)|} + \gamma \tag{5}$$

where $1/|L(t)|$ is the chance term and $\gamma$ is the significance term.

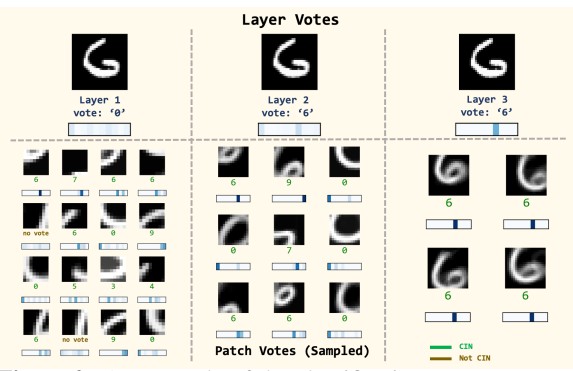

**Figure 2:** An example of the classification process. Every patch (at any layer) that selects a CIN centroid votes for the single class that has the highest association with. These patch votes are first averaged at each layer. The final inference is the class with the highest cumulative vote across all layers.

**III. Classification using a hierarchy of centroids:** At test time, we are given an input $\mathbf{x}$ of class $k(\mathbf{x})$ and infer its class as $\hat{k}(\mathbf{x})$. The classification task is a "biased voting" process in which every patch of $\mathbf{x}$, at any layer, votes for a single class as long as that patch selects a CIN centroid.

Specifically, if a patch $\mathbf{x_{l,m}}$ of layer $l$ selects a CIN centroid $\mathbf{c}$, then that patch votes $v_{l,m} = \max_{k\in L(t)} g_{\mathbf{c}}(k)$ for the class $k$ that has the highest association with $\mathbf{c}$, and zero for all other classes. If $c$ is *not* a CIN centroid, the vote of that patch is zero for all classes.

The vote of layer $l$ for class $k$ is the average vote across all patches in layer $l$ (as illustrated in Figure 2):

$$v_l(k) = \frac{\sum_{m\in M_l} v_{l,m}}{|M_l|} \tag{6}$$

where $M_l$ is the set of patches in layer $l$. The final inference for input $\mathbf{x}$ is the class with the highest cumulative vote across all layers:

$$\hat{k}(\mathbf{x}) = \arg\max_{k'} \sum_{l=1}^{\Lambda} v_l(k) \tag{7}$$

## 5    EVALUATION

To evaluate the STAM architecture in the UPL context, we consider a data stream in which small groups of classes appear in successive *phases*, referred to as **Incremental UPL**. New classes are

introduced two at a time in each phase, and they are only seen in that phase. STAM must be able to both recognize new classes when they are first seen in the stream, and to also remember all previously learned classes without catastrophic forgetting. Another evaluation scenario is **Uniform UPL**, where all classes appear with equal probability throughout the stream – the results for Uniform UPL are shown in SM-G. We include results on four datasets: MNIST (Lecun et al., 1998), EMNIST (balanced split with 47 classes) (Cohen et al., 2017), SVHN (Netzer et al., 2011), and CIFAR-10 (Krizhevsky et al., 2014). For each dataset we utilize the standard training and test splits. We preprocess the images by applying per-patch normalization (instead of image normalization), and SVHN is converted to grayscale. More information about preprocessing can be found in SM-H.

We create the training stream by randomly selecting, with equal probability, $N_p$ data examples from the classes seen during each phase. $N_p$ is set to 10000, 10000, 2000, and 10000 for MNIST, SVHN, EMNIST, and CIFAR-10 respectively. More information about the impact of the stream size can be found in SM-E. In each task, we average results over three different unlabeled data streams. During testing, we select 100 random examples of each class from the test dataset. This process is repeated five times for each training stream (i.e., a total of fifteen results per experiment). The following plots show mean $\pm$ std-dev.

For all datasets, we use a 3-layer STAM hierarchy. In the clustering task, we form the set $\Phi(\mathbf{x})$ considering only Layer-3 patches of the input $\mathbf{x}$. In the classification task, we select a small portion of the training dataset as the labeled examples that are available only to the classifier. The hyperparameter values are tabulated in SM-A. The robustness of the results with respect to these values is examined in SM-F.

**Baseline Methods:** We evaluate the STAM architecture comparing its performance to two state-of-the-art baselines for continual learning: GEM and MAS. We emphasize that there are no prior approaches which are directly applicable to UPL. However, we have taken reasonable steps to adapt these two baselines in the UPL setting. Please see SM-B for additional details about our adaptation of GEM and MAS.

**Gradient Episodic Memories (GEM)** is a recent supervised continual learing model that expects known task boundaries (Lopez-Paz & Ranzato, 2017). To turn GEM into an unsupervised model, we combined it with a self supervised method for rotation prediction (Gidaris et al., 2018). Additionally, we allow GEM to know the boundary between successive phases in the data stream. This makes the comparison with STAM somehow unfair, because STAM does not have access to this information. The results show however that STAM performs better even without knowing the temporal boundaries of successive phases.

**Memory Aware Synapse (MAS)** is another supervised continual learning model that expects known task boundaries (Aljundi et al., 2018). As in GEM, we combined MAS with a rotation prediction self-supervised task, and provided the model with information about the start of each new phase in the data stream.

To satisfy the stream requirement of UPL, the number of training epochs for both GEM and MAS is set to one. Deep learning methods become weaker in this streaming scenario because they cannot train iteratively over several epochs on the same dataset. For all baselines, the classification task is performed using a $K = 1$ Nearest-Neighbor (KNN) classifier – we have experimented with various values of $K$ and other single-pass classifiers, and report only the best performing results here. We have also compared the memory requirement of STAM (storing centroids at STM and LTM) with the memory requirement of the two baselines. The results of that comparison appear in SM-C.

**Clustering Task:** The results for the clustering task are given in Figure 3. Given that we have the same number of test vectors per class we utilize the *purity* measure for clustering accuracy. In MNIST, STAM performs consistently better than the two other models, and its accuracy stays almost constant throughout the stream, only dropping slightly in the final phase. In SVHN, STAM performs better than both deep learning baselines with the gap being much smaller in the final phase. In CIFAR-10 and EMNIST, on the other hand, we see similar performance between all three models. Again, we emphasize that STAM is not provided task boundary information while the baselines are and is still able to perform better, significantly in some cases.

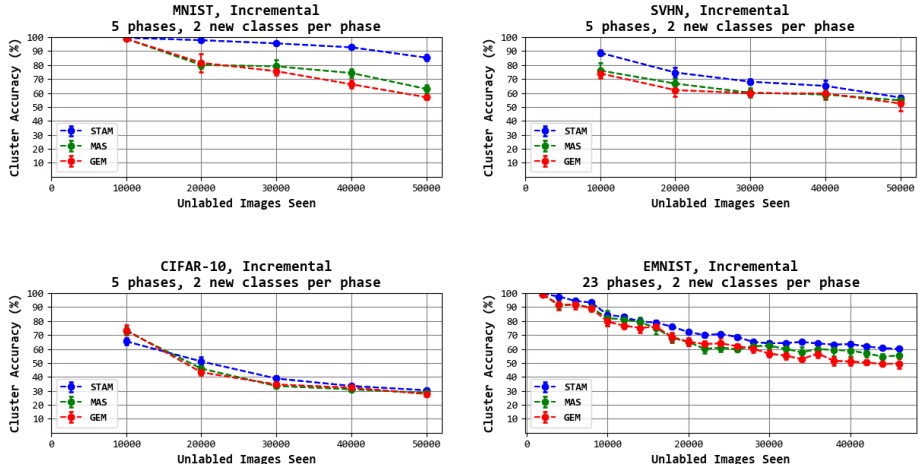

**Figure 3:** Clustering accuracy for MNIST (left), SVHN (left-center), CIFAR-10 (right-center), and EMNIST (right). The task is expanding clustering for incremental UPL. The number of clusters is equal to 2 times the number of classes in the data stream seen up to that point in time.

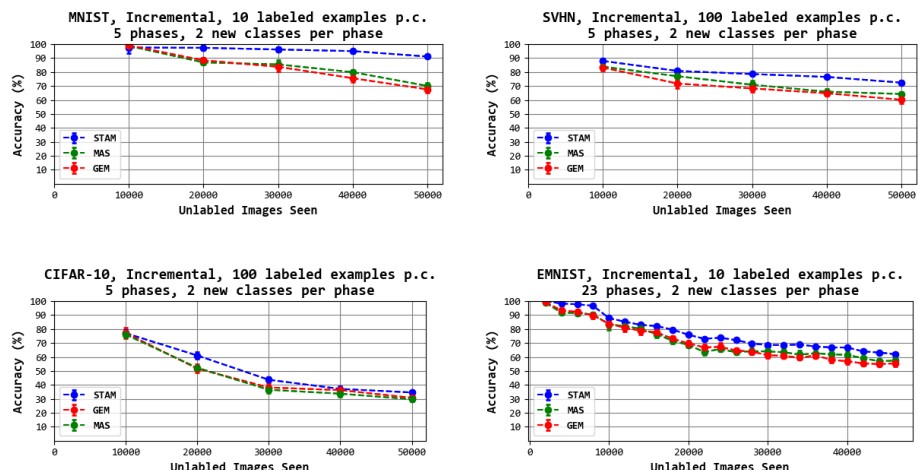

**Figure 4:** Classification accuracy for MNIST (left), SVHN (center), CIFAR-10 (right-center), and EMNIST (right). The task is expanding classification for incremental UPL, i.e., recognize all classes seen so far. Note that the number of labeled examples is 10 per class for MNIST and EMNIST and 100 per class for SVHN and CIFAR-10.

**Classification Task:**    We focus on an *expanding classification task*, meaning that in each phase we need to classify *all* classes seen so far. The results for the classification task are given in Figure 4. Note that we use only 10 labeled examples per class for MNIST and EMNIST, and 100 examples per class for SVHN and CIFAR-10. We emphasize that the two baselines, GEM and MAS, have access to the temporal boundaries between successive phases, while STAM does not.

As we introduce new classes in the stream, the average accuracy per phase decreases for all methods in each dataset. This is expected, as the task gets *more difficult* after each phase. In MNIST, STAM performs consistently better than GEM and MAS, and STAM is less vulnerable to catastrophic forgetting. For SVHN, the trend is similar after the first phase but the difference between STAM and both baselines is smaller. With CIFAR-10, we observe that all models including STAM perform rather poorly – probably due to the low resolution of these images. STAM is still able to maintain comparable accuracy to the baselines with a smaller memory footprint. Finally, in EMNIST, we see a consistently higher accuracy with STAM compared to the two baselines. We would like to emphasize that these baselines are allowed extra information in the form of known tasks boundaries (a label that marks when the class distribution is changing) and STAM is still performs better both on all datasets.

**A closer look at Incremental UPL:** As we introduce new classes to the incremental UPL stream, the architecture recognizes previously learned classes without any major degradation in classification accuracy (left column). The average accuracy per phase is decreasing, which is due to the increasingly difficult expanding classification task. For EMNIST, we only show the average accuracy because there are 47 total classes. In all datasets, we observe that layer-2 and layer-3 (corresponding to the largest two receptive fields) contain the highest fraction of CIN centroids (center column). The ability to recognize new classes is perhaps best visualized in the LTM centroid count (right column). During each phase the LTM count stabilizes until a sharp spike occurs at the start of the next phase when new classes are introduced. This reinforces the claim that the LTM pool of centroids (i) is stable when there are no new classes, and (ii) is able to recognize new classes via novelty detection when they appear.

In the CIFAR-10 experiment, the initial spike of centroids learned is sharp, followed by a gradual and weak increase in the subsequent phases. The per-class accuracy results show that STAM effectively forgets certain classes in subsequent phases (such as classes 2 and 3), suggesting that there is room for improvement in the novelty detection algorithm because the number of created LTM centroids was not sufficiently high.

In the EMNIST experiment, as the number of classes increases towards 47, we gradually see fewer "spikes" in the LTM centroids for the lower receptive fields, which is expected given the repetition of patterns at that small patch size. However, the highly CIN layers 2 and 3 continue to recognize new classes and create centroids, even when the last few classes are introduced.

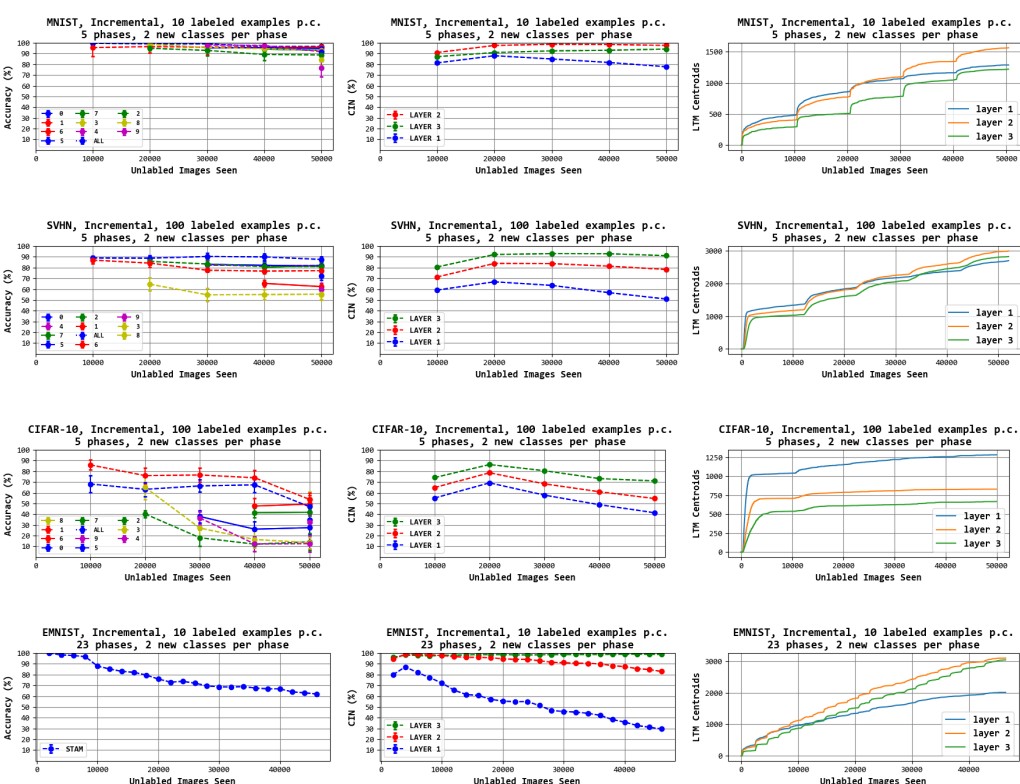

**Figure 5:** STAM Incremental UPL evaluation for MNIST (row-1), SVHN (row-2), EMNIST (row-3) and CIFAR-10 (row-4). Per-class and average classification accuracy (left); fraction of CIN centroids over time (center); number of LTM centroids over time (right). The task is expanding classification, i.e., recognize all classes seen so far.

**Ablation studies:** Several STAM ablations are presented in Figure 6. On the left, we remove the LTM capability and only use STM centroids for classification. During the first two phases, there is little (if any) difference in classification accuracy. However, we see a clear dropoff during phases 3-5. This suggests that, without the LTM mechanisms, features from classes that are no longer seen in the stream are forgotten over time, and STAM can only successfully classify classes that have been

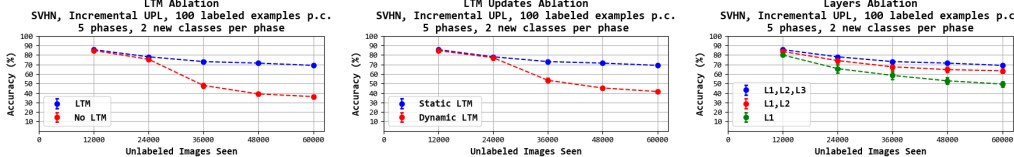

**Figure 6:** Ablation study: A STAM architecture without LTM (left), a STAM architecture in which the LTM centroids are adjusted with the same learning rate $\alpha$ as in STM (center), and a STAM architecture with removal of layers (right)

recently seen. We also investigate the importance of having static LTM centroids rather than dynamic centroids (Fig. 6-middle). Specifically, we replace the static LTM with a dynamic LTM in which the centroids are adjusted with the same learning rate parameter $\alpha$, as in STM. The accuracy suffers drastically because the introduction of new classes "takes over" LTM centroids of previously learned classes, after the latter are removed from the stream. Similar to the removal of LTM, we do not see the effects of "forgetting" until phases 3-5. Note that the degradation due to a dynamic LTM is less severe than that from removing LTM completely.

Finally, we look at the effects of removing layers from the STAM hierarchy (Fig. 6-right). We see a small drop in accuracy after removing layer 3, and a large drop in accuracy after also removing layer 2. The importance of having a deeper hierarchy would be more pronounced in datasets with higher-resolution images or videos, potentially showing multiple objects in the same frame. In such cases, CIN centroids can appear at any layer, starting from the lowest to the highest.

## 6 RELATED WORK

The UPL problem has some similarities with several recent approaches in the machine learning literature but it is also different in important aspects we describe in this section. Each paragraph highlights the most relevant prior work and explains how it is different from UPL.

**I: Continual learning:** In addition to CL models cited in the introduction, other *supervised* CL methods include regularization-based approaches (Aljundi et al., 2018; Golkar et al., 2019; Hayes & Kanan, 2019; Kirkpatrick et al., 2017; Yoon et al., 2018; Zenke et al., 2017), expanding architectures (Lomonaco & Maltoni, 2017; Maltoni & Lomonaco, 2019; Rusu et al., 2016), and distillation-based methods (Lee et al., 2019; 2020; Li & Hoiem, 2017). Their main difference with UPL and STAM is that they are designed for supervised learning, and it is not clear how to adapt them for non-stationary and unlabeled data streams.

**II. Offline unsupervised learning:** Additional *offline* representation learning methods include clustering (Caron et al., 2018; Jiang et al., 2017a; Xie et al., 2016; Yang et al., 2016), generative models (Eslami et al., 2016; Jiang et al., 2017b; Kosiorek et al., 2018; 2019), information theory (Hjelm et al., 2019; Ji et al., 2019), among others. These methods require prior information about the number of classes present in a given dataset (to set the number of cluster centroids or class outputs) and iterative training (i.e. data replay), and therefore cannot be directly applied in the UPL setting.

**III. Semi-supervised learning (SSL):** SSL methods require labeled data during the representation learning stage and so they are not compatible with UPL (Kingma et al., 2014; Lee, 2013; Miyato et al., 2018; Oliver et al., 2018; Rasmus et al., 2015; Springenberg, 2015; Tarvainen & Valpola, 2017).

**IV. Few-shot learning (FSL) and Meta-learning:** These methods recognize object classes not seen in the training set with only a single (or handful) of labeled examples (Fei-Fei et al., 2006; Finn et al., 2017; Ren et al., 2018; Snell et al., 2017). Similar to SSL, FSL methods require labeled data to learn representations and therefore are not applicable in the UPL context. Centroid networks (Huang et al., 2019) do not require labeled examples at inference time but require labeled examples for training. Few-Shot Incremental Learning (Ayub & Wagner, 2020) includes a similar evaluation protocol to our work, but includes pre-training and learns centroids using data labels.

**V. Multi-Task Learning (MTL):** Any MTL method that involves separate heads for different tasks is not compatible with UPL because task boundaries are not known a priori in UPL (Ruder, 2017). MTL methods that require pre-training on a large labeled dataset are also not applicable to UPL (Pan & Yang, 2010; Yosinski et al., 2014).

**VI. Online and Progressive Learning:** Many earlier methods learn in an online manner, meaning that data is processed in fixed batches and discarded afterwards. This includes progressive learning (Venkatesan & Er, 2016) and streaming with limited supervision (Chiotellis et al., 2018; Li et al., 2018; Loo & Marsono, 2015), both of which require labeled data in the training stream. (Caccia et al., 2019) is another very recent work which focuses on online continual compression using a discrete auto encoder and self-replay.

**VII. Continual Unsupervised Representation Learning (CURL):** Similar to the UPL problem, CURL (Rao et al., 2019) focuses on continual unsupervised learning from non-stationary data with unknown task boundaries. Like STAM, CURL also includes a mechanism to trigger dynamic capacity expansion as the data distribution changes. However, a major difference is that CURL is *not* a streaming method – it processes each training example multiple times. We have experimented with CURL but we found that its performance collapses in the UPL setting due to mostly two reasons: the single-pass through the data requirement of UPL, and the fact that we can have more than one new classes per phase. For these reasons, we choose not to compare STAM with CURL because such a comparison would not be fair for the latter.

**VIII. Data dimensionality and clustering-based representation learning**: As mentioned earlier in this section, clustering has been used successfully in the past for offline representation learning (e.g., (Coates et al., 2011; Coates & Ng, 2012)). Its effectiveness, however, gradually drops as the input dimensionality increases (Beyer et al., 1999; Hinneburg et al., 2000). In the STAM architecture, we avoid this issue by clustering smaller subvectors (patches) of the input data. If those subvectors are still of high dimensionality, another approach is to reduce the *intrinsic dimensionality* of the input data at each layer by reconstructing that input using representations (selected centroids) from the previous layer.

**VIII. Related work to other STAM components**: STAM relies on online clustering. This algorithm can be implemented with a rather simple recurrent neural network of excitatory and inhibitory spiking neurons, as shown recently (Pehlevan et al., 2017). The novelty detection component of STAM is related to the problem of anomaly detection in streaming data (Dasgupta et al., 2018) — and the simple algorithm currently in STAM can be replaced with more sophisticated methods (e.g., (Cui et al., 2016; Yong et al., 2012)). Finally, brain-inspired dual-memory systems have been proposed before for memory consolidation (e.g., (Kemker & Kanan, 2018; Parisi et al., 2018; Shin et al., 2017)).

## 7 DISCUSSION

The STAM architecture aims to address the following desiderata that is often associated with Lifelong Learning (Parisi et al., 2019):

**I. Online learning:** STAMs update the learned features with every observed example. There is no separate training stage for specific tasks, and inference can be performed in parallel with learning.

**II. Transfer learning:** The features learned by the STAM architecture in earlier phases can be also encountered in the data of future tasks (forward transfer). Additionally, new centroids committed to LTM can also be closer to data of earlier tasks (backward transfer).

**III. Resistance to catastrophic forgetting:** The STM-LTM memory hierarchy of the STAM architecture mitigates catastrophic forgetting by committing to "permanent storage" (LTM) features that have been often seen in the data during any time period of the training period.

**IV. Expanding learning capacity:** The unlimited capacity of LTM allows the system to gradually learn more features as it encounters new classes and tasks. The relatively small size of STM, on the other hand, forces the system to forget features that have not been recalled frequently enough after their creation.

**V. No direct access to previous experience:** STAM only needs to store data centroids in a hierarchy of increasing receptive fields – there is no need to store previous exemplars or to learn a generative model that can produce such examples.

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
