# OpenReview forum: "Unsupervised Progressive Learning and the STAM Architecture"
_ICLR.cc/2021/Conference — Reject_

### Official Review · AnonReviewer3 · 2020-10-28
**I'm not sure if the evaluations are fair**

**Rating:** 5
**Confidence:** 3

**Review:**

##########################################################################
Summary:

The paper introduces a non-parametric approach, STAM, for unsupervised progressive learning (UPL), a variant of continual unsupervised learning with a single-stream requirement. STAM is developed for visual tasks. It comprises several components: (1) online clustering of hierarchical visual features (2) novelty detection (3) dual-memory for prototypical features. Experiments show STAM performs better than GEM, MAS in specific scenarios.

##########################################################################

Reasons for score:

Currently, I vote for rejecting the paper. The paper is well written, easy to follow. My main concerns are (1) to what extent UPL is a real and practical problem, compared with unsupervised continual learning; (2) the comparisons with GEM, and MAS do not look fair; (3) each technical component is not new.

##########################################################################

Pros:

The paper is well written, with enough details (and code) to reproduce the results;

The proposed method is technically sound;

Discussions on related work are comprehensive;

The experiments are thorough. Ablations are performed to justify the various design considerations.

##########################################################################

Cons:

I’m not sure if UPL is a real and practical problem. To me, the single-stream requirement looks somewhat unnecessary and is unfair to all parametric models (including deeply learned models), especially when the number of training iterations is small (e.g. on epoch on MNIST). On the other hand, as STAM relies on prototypical features from online clustering, essentially no parameters need to be learned. Thus, the comparisons between STAM and MAS/GEM presented in the paper are unfair as I believe the MAS/GEM models were highly undertrained in this case. What if STAM/MAS/GEM are all trained for two epochs, or trained on a larger dataset for one epoch? I guess the performance of MAS/GEM will be improved but I’m not sure if the performance of STAM can be improved.

STAM is only evaluated on small datasets (e.g. MNIST, SVHN, CIFAR10). I’m not sure if it is expressive enough to model larger datasets (e.g. CIFAR 100) with a reasonable memory footprint (e.g. comparable with a deeply learned model)

---

> ### Author Response · Authors · 2020-11-23
> **Response to Reviewer 3**
>
> Thank you for the constructive comments. Please find our responses to your individual comments below.
>
>
> * to what extent UPL is a real and practical problem, compared with unsupervised continual learning.
>
> **Response:**
> We kindly invite the reviewer to consider applications in which the model must perform both learning and inference in real-time. For instance an exploratory robot, a self-driving vehicle, or a drone -- all of them operating in a dynamic environment in which they will need to constantly learn new classes of objects and new tasks -- but under the pragmatic constraint that the learning should be performed in real-time, processing visual (or even multimodal) new data before the next frame of data is generated (for video this means processing a new frame within about 30 msec).
> Many of the recently proposed DL architectures for continual learning require training over multiple epochs on powerful GPU systems. Our goal in designing this, admittedly much simpler, architecture was to create a solution that can easily run in real-time even in under-powered systems. This is why we use online clustering instead of more powerful embedding operations -- it can be performed in a single pass over the data.
>
> * the comparisons with GEM, and MAS do not look fair;
>
> **Response:**
> We agree with the reviewer that GEM, MAS (and CURL) were not developed for the single-pass UPL context. But please also consider the following issue.
> In an earlier version of our paper, we had made the case that there are no published baselines in the literature that address the single-pass UPL problem -- and for that reason we chose to not compare STAM with any published deep learning baselines in that paper. Instead we compared STAM with a couple of deep learning schemes that we developed as baselines (one based on a convolutional autoencoder and another based on a rotation-based self-supervised model). That earlier version of the paper was rejected from another major ML conference however, mostly because we did not include comparisons with “state-of-the-art deep learning baselines”.
> So we felt the need to include comparisons with GEM/MAS/CURL, after adopting them to the UPL context as well as we could. We even gave those methods an advantage over STAM, by providing them with information about the end/start of each new phase (temporal boundary between learning new classes).
> Following this reviewer’s suggestion, we even tried the option that GEM and MAS are trained for two epochs instead of only one -- but their accuracy did not improve significantly.
> At this point we do not know what else to do -- some reviewers are asking us to provide comparisons with SOTA baselines while others think that these comparisons are unfair. We welcome any suggestions from the ICLR reviewers, even if our paper is rejected.
>
> * STAM is only evaluated on small datasets (e.g. MNIST, SVHN, CIFAR10). I’m not sure if it is expressive enough to model larger datasets (e.g. CIFAR 100) with a reasonable memory footprint (e.g. comparable with a deeply learned model)
>
> **Response:**
>  We also show results on EMNIST which has 47 classes. First, as Figure-5 shows (rightmost column -- one plot for each dataset), the number of LTM centroids increases sublinearly. For instance, the number of LTM centroids learned at the last training phase, relative to the number of LTM centroids learned in the first training phase, is: 62% for MNIST, 25% for EMNIST, 34% for SVHN, and 2% for CIFAR-10. So, as the model observes more classes, it needs to learn fewer and fewer new centroids per class. This is because centroids learned from earlier classes are often able to represent features of new classes. Of course, every new class will have its own discriminative features  - and this will be causing a slow increase in the number of LTM centroids as the model learns more classes.
> The second part of our answer is related to this last point. We think that it is reasonable to expect that the capacity of a continual learning model (including STAM) should increase over time, as the model learns more and more classes. If the capacity is finite (say we limit the LTM buffer size), it is reasonable to expect that the model will not be able to learn new classes after a certain point without catastrophic forgetting.

---

### Official Review · AnonReviewer2 · 2020-10-28
**The authors propose an approach (architecture + algorithms) to unsupervised progressive learning in a non-stationary environment (the number of classes grows gradually) by keeping centroids at several hierarchies, using a combination of techniques from online clustering, via computing and updating centroids, with novelty detection, and dropping (forgetting those deemed outliers).**

**Rating:** 6
**Confidence:** 3

**Review:**

The authors propose an approach (architecture + algorithms) to unsupervised progressive learning in a non-stationary environment (the number of classes grows gradually) by keeping centroids at several hierarchies, using a combination of techniques from online clustering, via computing and updating centroids, with novelty detection, and dropping (forgetting those deemed outliers).  A variety of experiments are performed on several image datasets (MNIST, EMNIST, SVHN, CIFAR-10) with comparisons to other adapted methods. They evaluate performance in a supervised setting where they describe how they learn centroid to label(s) mappings.



The experiments are somewhat promising,  and I liked the hierarchical aspect of the centroids.

But I have concerns about
applicability of the approach to practice: The approach has some of
the shortcomings of nearest neighbors: efficiency specially at
classification/test time (as the number of classes goes up to many
1000s, and required number of centroids per class goes up), choice of
distance, and so on.     What do you do if too many centroids?  space and
speed of finding closest centroids to a patch?  How is fast nearest
neighbor done?  Also, different classes require different complexities
(in terms of distance to use, manifold learning), and one parameter
for novelty detection and another for when to add a centroid to LTM,
may not be sufficient...

Regarding evaluation: Why the particular elaborate learning/classification scheme chosen (that computes
 a mapping from centroids to labels)?    the research or
motivation behind it? Why not a plain classifier, such as a simple nn,  svm, etc... given the features (centroids)
available? Perhaps because there is no explicit feature representation.    A quick discussion would be good.


The paper was overall clearly written, and the supplements provide much useful detail on the experiments.

Some  detailed comments:

section I, hierarchy: ruo_l x ruo_l, the subscript 'l' is not
explained  before introduced (later as layer). It would be good to
quickly give examples of the patch dimensions.

section 2.II:

what is t in C_l(t) (in footnote, they say they drop time index, so t
is probably time)

Does the online update take place only if (patch is) not deemed
outlier? The text seems to imply that, but it's not clear.  How do you
(re)use centroids that have been moved to long-term memory? It's not
clear from the text but it appears all centroids are used to compute
the nearest centroid (both in LTM and STM, ie C_l is the union).

---

> ### Author Response · Authors · 2020-11-23
> **Response to Reviewer 2**
>
> Thank you for the constructive comments. Please find our responses to your individual comments below.
>
> * The approach has some of the shortcomings of nearest neighbors: efficiency especially at classification/test time (as the number of classes goes up to many 1000s, and required number of centroids per class goes up), choice of distance, and so on. What do you do if too many centroids?
>
> **Response:**
> The reviewer’s concern is one of our future work areas. We have not evaluated the STAM architecture yet in datasets with thousands of classes or millions of centroids. It is likely we will need to make architectural changes so that the model can remain computationally efficient and able to perform learning and inference in real-time.
> We think of this first paper as an “introduction to the UPL problem”, together with a simple but effective architecture to address this problem in the case of the most common datasets and tasks that the ML community often starts from.
>
> * space and speed of finding closest centroids to a patch? How is fast nearest neighbor done?
>
> **Response:**
> Regarding the memory requirements of STAM, we invite the reviewer to also read the Supp-Material section (C), where we compare the memory footprint of STM (including both STM and LTM) with some deep learning models.
> Regarding the computation of nearest-neighbor centroids: suppose that each input consists of $p$ patches, and each patch is of length $k$. Also, suppose that the LTM stores $m$ centroids. The computation of the Euclidean distance between a patch and a centroid takes $O(k)$. We compute this distance between every patch with every centroid. So the total computation for each input is $O(p \times k \times m)$. For a given dataset, k and p will be constants, and so the nearest-neighbor computation is linear with the number of centroids.
> In terms of actual implementation, we STAM is written in Python and we have been using the highly optimized vector and matrix operations library available in NumPy. We have recently improved this process using Numba to optionally calculate this pairwise distance matrix on CUDA-enabled GPU systems.
>
> *  Also, different classes require different complexities (in terms of distance to use, manifold learning), and one parameter for novelty detection and another for when to add a centroid to LTM, may not be sufficient…
>
> **Response:**
> We do not disagree with the reviewer -- but please also consider our response to your first point above.
>
> * Why the particular elaborate learning/classification scheme chosen (that computes a mapping from centroids to labels)? the research or motivation behind it? Why not a plain classifier, such as a simple nn, svm, etc... given the features (centroids) available? Perhaps because there is no explicit feature representation. A quick discussion would be good.
>
> **Response:**
> We have experimented with variations of KNN classifiers and the proposed classifier performs better with the STAM centroids. We avoided some other classifiers, such as SVM or a neural network, because of the single-pass training requirement.
>
>
> * section I, hierarchy: ruo_l x ruo_l, the subscript 'l' is not explained before introduced (later as layer). It would be good to quickly give examples of the patch dimensions.
>
> **Response:**
> Thank you -- we have clarified and also given an example.
>
> * what is t in C_l(t) (in footnote, they say they drop time index, so t is probably time)
>
> **Response:**
> Thank you -- we have clarified and also given an example.
>
> * Does the online update take place only if (patch is) not deemed outlier? The text seems to imply that, but it's not clear. How do you (re)use centroids that have been moved to long-term memory? It's not clear from the text but it appears all centroids are used to compute the nearest centroid (both in LTM and STM, ie $C_l$ is the union).
>
> **Response:**
> Your interpretation is correct. A patch leads to the update of an STM centroid if that patch is not deemed an outlier (LTM centroids, on the other hand, are not updated).
> Also, as you mentioned, we compare a patch with the entire centroid pool (both in STM and LTM).

---

### Official Review · AnonReviewer4 · 2020-10-29
**Lays the foundation for a needed new problem setting for unsupervised progressing learning that acceptably addresses limitations of similar settings..**

**Rating:** 7
**Confidence:** 3

**Review:**

The authors propose and formulate a new problem setting UPL which addresses a number of limitations of current methods. I think this problem formulation is very important and relevant and the authors provide a fairly solid foundation about how to address this problem.

There are a number of strengths to this work. I found the non-deep learning approach they took refreshing and their experimentation demonstrates the effectiveness of it in this setting. I found the evaluation relatively convincing. More specifically, I find the classification and clustering results good. Again, the classification results are difficult to interpret as you do use the labels (albeit not for learning the representation), but it becomes more like semi-supervised learning at this point. On the other hand, the clustering results do not have this problem and are more impressive. I appreciate the code being included in the supplementary material.

I have a number of questions/concerns about the work.
	1) How effective is this hierarchical receptive field processing approach? My feelings are that it may be insufficient for larger/more complex data. Evaluation involving these would be much welcomed. Further, how does it perform for non-image data types?
	2) The focus on classification is a little out of place, I can see why it's useful, but as a main section of the paper, preceding the purely unsupervised setting, it can be confusing to readers. It would be clearer in my opinion if the focus was on the unsupervised setup, and then the semi-supervised setup as an additional evaluation (i.e., not preceding the purely unsupervised approach as it does now). This is a criticism of the structure of the paper (and perhaps the main message), more than the work itself.
	3) It seems like a limitation of this approach is that clustering algorithms that can be used on the representation are limited to discrete distance measures. Can the authors speculate on potential future limitations of this for the setting?
	4) "the small batch size required by UPL" I do not follow this - can you further explain?
	5) Further, while it may be unfair on CURL to compare directly, but as it is a very similar approach,  I do think it would be interesting to see it how it performs, while acknowledging the difficulties and reasons why. If not a full set of experiments, "We have experimented with CURL but we found that its performance collapses in the UPL setting" leaves me wondering what exactly the collapse is.

Overall, I think this is worthy of acceptance, acknowledging the fact  that it is early work in this new area.

---

> ### Author Response · Authors · 2020-11-23
> **Response to Reviewer 4**
>
> Thank you for the constructive comments. Please find our responses to your individual comments below.
>
> * How effective is this hierarchical receptive field processing approach? My feelings are that it may be insufficient for larger/more complex data. Evaluation involving these would be much welcomed. Further, how does it perform for non-image data types?
>
> **Response:**
> We have not worked so far with any non-image data -- even though it is certainly in our “to-do” list, and we do not foresee major changes in the architecture/model.
> One challenge when we move to larger or more complex images (e.g., images with multiple objects or many distractors) is that some form of indirect supervision (or domain knowledge) will be necessary in the selection of the receptive field dimensions at each layer. For instance, are the important features in a given visual feed expected to reside in 8x8 patches or 64x64 patches?
>
> * The focus on classification is a little out of place, I can see why it's useful, but as a main section of the paper, preceding the purely unsupervised setting, it can be confusing to readers. It would be clearer in my opinion if the focus was on the unsupervised setup, and then the semi-supervised setup as an additional evaluation (i.e., not preceding the purely unsupervised approach as it does now). This is a criticism of the structure of the paper (and perhaps the main message), more than the work itself.
>
> **Response:**
> We agree and have revised the paper to introduce the clustering discussion and results before the classification part.
>
> * It seems like a limitation of this approach is that clustering algorithms that can be used on the representation are limited to discrete distance measures. Can the authors speculate on potential future limitations of this for the setting?
>
> **Response:**
> We admit that we do not understand this question. Is it possible that the reviewer explains a bit more the setting he/she refers to?
>
> * "the small batch size required by UPL" I do not follow this - can you further explain?
>
> **Response:**
> We rewrote that part to avoid any confusion: instead of “small batch size” we now refer to the “single pass through the data UPL requirement.”
>
> * Further, while it may be unfair on CURL to compare directly, but as it is a very similar approach, I do think it would be interesting to see it how it performs, while acknowledging the difficulties and reasons why. If not a full set of experiments, "We have experimented with CURL but we found that its performance collapses in the UPL setting" leaves me wondering what exactly the collapse is.
>
> **Response:**
> The final classification accuracy of CURL on MNIST in the UPL setting is roughly 61% (compared to roughly 92% for STAM), and the clustering accuracy is roughly 52% (compared to roughly 87% for STAM). We did not get performance above chance for the other datasets.
> We do not report these results in the main paper because we are using CURL outside of its intended setting, and thus the performance is quite poor. For example, CURL does not work with small batch sizes because it relies on a categorical regularization prior to encourage data from each batch to be evenly distributed amongst its mixture components (i.e., clusters). According to the original paper, the optimal hyperparameters for CURL result in roughly 20 clusters, meaning that each batch of data should be evenly distributed amongst the 20 clusters; thus, the batch size should be much greater than 20. However, a large batch size in the single-pass setting results in fewer training steps (i.e., gradient updates). This negatively affects the other components of CURL, mainly the expansion process. We found with a quick reasonable search of the hyperparameters that either: 1) expansion is too slow and not all classes are captured, or 2) expansion is too quick and the clusters are ill-formed, resulting in the majority of new images being considered outliers rather than matching to existing clusters.
> In summary, we found that CURL requires work beyond a simple hyperparameter search to be adapted for the UPL problem in a fair and meaningful way. We would like to emphasize that it is certainly possible that CURL could be modified and tuned to perform better, but we feel it is unlikely that it would be able to exceed STAM’s performance.

---

### Official Review · AnonReviewer1 · 2020-11-03
**Inappropriate evaluation setup - more engineering than science**

**Rating:** 2
**Confidence:** 5

**Review:**

In this paper, it is proposed to have a new problem setting of "unsupervised feature learning", which as claimed, is supposed to be different from standard continual learning settings. In this proposed setting, a data point can be seen only once by a model for training. I think, this is too strict a condition, as it is reasonable for a model to keep a data point in memory for a short while so as to use it for training across multiple epochs. In the experimental setup, neural baselines are trained only for a "single" epoch, in consideration of the proposed problem setting, which doesn't make sense for all practical purposes as neural models need a decent number of epochs for training. Furthermore, the final classifier used after learning the unsupervised representations is k-NN which is again unrealistic in the context of continual learning literature, especially in the context of neural baselines. All of this is justified for accommodating the  proposed problem setting which is itself not well motivated.

The proposed model for the introduced problem setting is more of an engineering approach, relying upon some basic techniques such as clustering, novelty detection. There is no clear motivation for the learning algorithm, it is not clear how the model is optimized wholistically. It is more of a heuristic driven approach. The model is claimed to be brain-inspired; for instance there is a component in the proposed model which has a "hierarchy of increasing receptive fields", which is nothing but CNN-like neural net.

For experiments, I have the following suggestions.
(1) In Fig. 3, x-label should be changed to something more appropriate, number of new classes introduced.
(2) Results are good for MNIST dataset primarily.
(3) Any other evaluation metrics besides accuracy? Accuracy can be misleading for multi-class scenario.
(4) Report accuracies separately for new classes introduced and the old classes.
(5) It needs to be clarified exactly what "class boundary information" the baseline methods have access to.
(6) Only 10 new classes introduced from the 5 phases, what about the datasets with a large number of classes?

---

> ### Author Response · Authors · 2020-11-23
> **Response to Reviewer 1**
>
> Thank you for the constructive comments. Please find our responses to your individual comments below.
>
> * In this proposed setting, a data point can be seen only once by a model for training. I think, this is too strict a condition, as it is reasonable for a model to keep a data point in memory for a short while so as to use it for training across multiple epochs.
>
> **Response:**
> We kindly invite the reviewer to consider applications in which the model must perform both learning and inference in real-time. For instance an exploratory robot, a self-driving vehicle, or a drone -- all of them operating in a dynamic environment in which they will need to constantly learn new classes of objects and new tasks -- but under the pragmatic constraint that the learning should be performed in real-time, processing visual (or even multimodal) new data before the next frame of data is generated (for video this means processing a new frame within about 30 msec).
> Many of the recently proposed DL architectures for continual learning require training over multiple epochs on powerful GPU systems. Our goal in designing this, admittedly much simpler, architecture was to create a solution that can easily run in real-time even in under-powered systems. This is why we use online clustering instead of more powerful embedding operations -- it can be performed in a single pass over the data.
>
> * The proposed model for the introduced problem setting is more of an engineering approach, relying upon some basic techniques such as clustering, novelty detection. There is no clear motivation for the learning algorithm, it is not clear how the model is optimized wholistically.
>
> **Response:**
> We respectfully disagree with the reviewer on this point. Even though our paper does not make a new theoretical contribution, it applies existing components on a new problem -- and we make the case that this problem is important in some applications in practice. Also, we hope that the publication of this work will trigger more theoretical follow-up work on the UPL problem.
> Additionally, we motivated the approach mostly focusing on its resemblance to biological learning -- we can expand the motivation if the reviewer thinks that that would be beneficial.
> Also, STAM is based on a well-defined local optimization at each layer (online clustering). We deliberately avoided solutions that require end-to-end optimization because they typically require multiple iterations.
>
> * In Fig. 3, x-label should be changed to something more appropriate, number of new classes introduced.
>
> **Response:**
> We considered and tried this visualization approach -- but we felt eventually that it is better to show the actual number of “examples seen” at the x-axis. The reader can easily understand the number of new classes in each phase (it is also mentioned at the title of the plot), and it is important (we think) to also show how many examples are used in each training phase.
>
> * Any other evaluation metrics besides accuracy? Accuracy can be misleading for multi-class scenario.
>
> **Response:**
> Please note that we show per-class accuracy results in what now appears as Figure-5.
>
> * Report accuracies separately for new classes introduced and the old classes.
>
> **Response:**
> Please see previous response -- we think that reporting per-class classification accuracy is the most informative metric.
> * It needs to be clarified exactly what "class boundary information" the baseline methods have access to.
>
> **Response:**
> During training, STAM is not provided with any information about the temporal boundary of phases (when we start seeing examples of new classes).  Both baselines are given this information during training. So, GEM can partition the stored examples by task, and MAS can calculate its important “omega values” properly at the end of each task.
>
> * Only 10 new classes introduced from the 5 phases, what about the datasets with a large number of classes?
>
> **Response:**
> While three of the four datasets we evaluate on have only ten classes, we also show results for the balanced 47-class version of EMNIST. We plan to work with additional datasets, with 100s or 1000s of classes in future work.

---

### Official Review · AnonReviewer5 · 2020-11-06
**Review: Unsupervised Progressive Learning and the STAM Architecture**

**Rating:** 5
**Confidence:** 3

**Review:**

This paper presents an "Unsupervised progressive learning" (UPL) problem, where a model is exposed to data in an non-iid manner, and each training example is presented once. Simple to continual learning, but a little more explicit in the connections to the way biological agents learn. They present a model that uses clustering and long-term memory (buffered) and compare on a few UPL tasks with additional supervision signal (classification) or unsupervised (clustering).

The paper is interesting. I appreciate the straightforward outline of the problem and connection to biological learning and some of the motivations for the model (e.g., long term buffered memory of centroids).

The model itself though isn't very clear in a few regards that I think are rather important.

1) for the architecture itself, are the cluster centroids propagated to the next layer, or is the same input used at the next layer? If so, I'm not sure what is hierarchal here, it's just ordered in terms of "patch dimension".

2) How large does the LTM get during training? Is there a maximum size? I didn't see any restrictions on the size of the LTM, so it seems this buffer could get more and more precise and eventually resemble just a normal buffer over remembered examples (with some small permutations depending on alpha.

3) During supervision the model has access to the number of classes?

Other questions:
Instead of a separate test-set, one could evaluate online, collecting the classification accuracies across learning before the learning step in the centroids, since each example is seem exactly once.

My main concerns though are on the size of the LTM and if this is really doing anything other than KNN with K=1 over a buffered memory of examples. I need to have some evidence that the LTM is doing something *general* w.r.t. those cluster centroids.

---

> ### Author Response · Authors · 2020-11-23
> **Response to Reviewer 5**
>
> Thank you for the constructive comments. Please find our responses to your individual comments below.
>
> * For the architecture itself, are the cluster centroids propagated to the next layer, or is the same input used at the next layer? If so, I'm not sure what is hierarchical here, it's just ordered in terms of "patch dimension".
>
> **Response:**
> The reviewer is right that strictly speaking the architecture is not hierarchical -- in the sense that the output of a layer is not fed to the input of the next layer. It is hierarchical in the sense that different layers operate on inputs of different dimensionality, allowing the architecture to identify features at different spatial scales.
>
> * How large does the LTM get during training? Is there a maximum size? I didn't see any restrictions on the size of the LTM, so it seems this buffer could get more and more precise and eventually resemble just a normal buffer over remembered examples (with some small permutations depending on alpha.
>
> **Response:**
> There are two parts in this response.
> First, as Figure-5 shows (rightmost column -- one plot for each dataset), the number of LTM centroids increases sublinearly. For instance, the number of LTM centroids learned at the last training phase, relative to the number of LTM centroids learned in the first training phase, is: 62% for MNIST, 25% for EMNIST, 34% for SVHN, and 2% for CIFAR-10.  So, as the model observes more classes, it needs to learn fewer and fewer new centroids per class. This is because centroids learned from earlier classes are often able to represent features of new classes. Of course, every new class will have its own discriminative features  - and this will be causing a slow increase in the number of LTM centroids as the model learns more classes.
> The second part of our answer is related to this last point. We think that it is reasonable to expect that the capacity of a continual learning model (including STAM) should increase over time, as the model learns more and more classes. If the capacity is finite (say we limit the LTM buffer size), it is reasonable to expect that the model will not be able to learn new classes after a certain point without catastrophic forgetting.
>
> * During supervision the model has access to the number of classes?
>
> **Response:**
> Yes, the classifier has access to the number of classes -- as it is given a number of labeled examples per class.
>
> * Instead of a separate test-set, one could evaluate online, collecting the classification accuracies across learning before the learning step in the centroids, since each example is seen exactly once.
>
> **Response:**
> This is a very good idea and we are planning to work on this. Unfortunately we were not able to generate these new results during the rebuttal period.
>
> * My main concerns though are on the size of the LTM and if this is really doing anything other than KNN with K=1 over a buffered memory of examples. I need to have some evidence that the LTM is doing something general w.r.t. those cluster centroids.
>
> **Response:**
> The reviewer is asking an important question: do the learned LTM centroids have good generalization ability (they are common features across different examples of the same class) as well as discriminative ability (they are distinct features for each class).
>
> Please note that we already have a metric in the paper that evaluates the association of a given LTM centroid with each class, based on a set of labeled examples of each class. Please see Equation (4). For centroid $c$, and for class $k$, the term $g_c(k)$ is the association between centroid $c$ and class-$k$, a number between 0 and 1. The closer that metric is to 1, the better that centroid is in terms of its ability to generalize across examples of class-k and to discriminate examples of that class from other classes.
>
> To address the reviewer’s question, we have performed the following experiment: we have calculated for each STAM centroid the maximum g value across all classes. This gives us a distribution of “max-g” values. We compare that distribution with a null model in which we have the same number of LTM centroids -- but those centroids are randomly chosen patches from the training dataset. These results are shown in a new subsection at the Supp-Material (SM-D). We also compare the two distributions (STAM versus “random examples”) using the Kolmogorov-Smirnov test -- the distributions are clearly different (the p-values are extremely small) and the STAM centroids have higher max-g values than the random examples. Of course there is still room for improvement, especially for CIFAR-10, to learn even better features -- meaning LTM centroids with higher max-g values.

---

### Decision · Program_Chairs · 2021-01-07
**Final Decision**

**Decision:**

Reject

**Comment:**

After reading the reviews, rebuttal, and looking through the paper I do feel that UPL setting is one that we need to consider. However is not clear to me that proposed approach matches the conditions described by the authors. In particular the scalability constraints seem important. I do feel that for the UPL setting makes sense particularly in the large case scenario of many examples and classes and how the system behaves under strict computational budgets for learning and inference. And I'm wondering whether in that limit parametric models would actually becomes relevant again, and whether there is a "burn-in" that one has to pay to use parametric models.
That said I don't think current approaches (CURL, AGEM etc.) will do well even in that setting, partially because they were not necessarily thought for that.
So in summary, I find the problem interesting, probably more so than the solution and particularly in an large scale setting.

However I think for the paper to have the impact it needs, and be ready for acceptance it needs a bit more. I think looking at a larger scale setting, and relying on that to motivate the problem will considerably help with its impact.  Also is not clear to me how the proposed solution scales (non-parametric approaches don't always do well in large scale settings), which I think is needed for it to be convincing.